# Quantitative Assessment of Impact of the Proposed Poyang Lake Hydraulic Project (China) on the Habitat Suitability of Migratory Birds

**Siyang Yao [1], Xinyu Li [1], Chenglin Liu [1,2,\*], Dongyang Yuan [1], Longhui Zhu [1], Xiangyu Ma [1], Jie Yu [1], Gang Wang [1] and Weiming Kuang [3]**

1  School of Civil Engineering and Architecture, Nanchang University, Nanchang 330031, China
2  Key Laboratory of Poyang Lake Environment and Resource Utilization, Nanchang University, Ministry of Education, Nanchang 330031, China
3  Jiangxi Water Resources Institute, Nanchang 330013, China
\*  Correspondence: liucl@ncu.edu.cn; Tel.: +86-135-1700-8889

**Abstract:** Poyang Lake is the largest wintering habitat for migratory birds in Asia. In the last decade, the lake has experienced an early-occurring and prolonged dry season that has deteriorated the lake's ecological status. To tackle this issue, the Chinese government has proposed the construction of the Poyang Lake Hydraulic Project (PLHP) to regulate water flow to the lake. However, its impact on migratory bird habitats is unknown. In this study, we simulated the habitat suitability for migratory birds in Poyang Lake during wet and dry years, with and without the presence/operation of the hydraulic project. A two-dimensional hydrodynamic model was used to simulate the water conditions for each case. Matter-element theory, 3S technology and ecological knowledge were combined to develop a matter-element-based habitat suitability model in a geographic information system (GIS)-based platform. We assessed and compared the habitat suitability in four scenarios: (1) Wet year without the hydraulic project, (2) wet year with the hydraulic project, (3) dry year without the hydraulic project, and (4) dry year with the hydraulic project. The results showed that the operation of the hydraulic project can effectively alleviate the water shortage issue in the wetland and increase the area of habitats suitable for migratory birds in typical dry years. However, it can reduce the area of suitable habitats in the northern provincial nature reserve of the lake. In addition, a reasonable management of the lake's fishing activities can also increase habitat suitability and promote balanced patterns between human activities and migratory bird habitats.

**Keywords:** habitat suitability; matter-element theory; the Poyang Lake Hydraulic Project; numerical simulation

## 1. Introduction

Worldwide, wetlands have been subjected to increasing human pressures which degrade their "ecological condition" [1–3]. Migratory birds are indicator organisms for the ecological status of wetlands that are heavily affected by human activities [4,5]. In the early 1980s, the world's largest *Grus leucogeranus* population was found to be wintering in the wetland of Poyang Lake. Poyang Lake and its wintering migratory birds have since attracted the attention of the international community [6]. Nowadays, more than 300,000 birds, including more than 30 rare species, visit Poyang Lake. Thus, Poyang Lake has the reputation of "the migratory bird kingdom" [7]. Since the beginning of the 21st century, especially after the operation of the Three Gorges Project in 2003, the dry season in Poyang Lake has occurred earlier and has lasted longer than normal [8,9]. This phenomenon results in an earlier occurrence and longer exposure time of the lake's beaches, causing a decline in the aquatic vegetation

upon which some waterbirds feed, thereby reducing their effective foraging area. In response to this phenomenon, the Jiangxi Government, in June 2013, proposed the construction of a hydraulic project in Poyang Lake, working with the concept that "the project should benefit both rivers and lakes" and that "the project should only control the flow during the dry season" to improve the water resources and increase the carrying capacity of Poyang Lake during the dry season. The Poyang Lake Hydraulic Project (PLHP) has entered the stage of feasibility study. The potential impacts of the hydraulic project on the wetland's ecology and its associated bird community have attracted international attention [10]. The hydraulic project will affect the inflows and outflows of the lake [9], which will influence the habitat of wintering migratory birds in Poyang Lake. The PLHP's impact on migratory bird habitat is a key determinant of whether this project can be implemented, and, thus, it should be systematically studied. The impact of the PLHP on migratory bird habitat suitability has been previously qualitatively analyzed by Guo et al. [11] and unilaterally (using only water level) by Wang et al. [12] and Lai et al. [13]. However, quantitative assessments are currently missing.

The use of 3S technology [14] to monitor and evaluate the habitat quality of migratory birds has gradually formed a research and development direction. Shealer et al. [15] ranked the suitability of US wetlands in 2010 by quantifying the habitat characteristics of *Chlidonias niger*. The conclusions were highly consistent with the results obtained by using the two methods of remote sensing and field survey. Van Schmidt et al. [16] developed habitat suitability maps by selecting habitat composition values to determine optimal habitat for *Grus americana* in 2011. Tang et al. [17] selected evaluation factors according to the habits of wintering *Anseriformes* and used the Habitat Suitability Index (HSI) to establish a suitability evaluation model for wintering *Anseriformes*' habitat. Therefore, migratory bird habitat suitability can be quantitatively studied by building a habitat suitability evaluation model. However, it is difficult to establish a comprehensive habitat evaluation model after the completion of the PLHP, because the water regime (discharge, water level, flow velocity, water surface, etc.) is currently unknown. Poyang Lake's water regime cannot be accurately simulated by a simple model because it has a complex shoreline and a highly variable terrain with considerable seasonal flow variability [18]. With the development of numerical simulation technology, it is possible to obtain some previously unavailable data [18,19]. Therefore, numerical simulation technology can be used to simulate the basic water regime after the completion of the PLHP and extracted required data.

In this study, we developed a model in combination with 3S technology and a two-dimensional hydrodynamic model to evaluate and simulate the habitat suitability for migratory birds in Poyang Lake before and after the establishment and operation of the PLHP. We modeled the habitat suitability during wet and dry years, with and without the presence of the PLHP, based on the latest dispatch plan proposed by the Poyang Lake Water Conservancy Construction Office in May 2017. Habitat suitability was evaluated for four scenarios: 1: Wet year without the PLHP (S1); 2: Dry year without the PLHP (S2); 3: Wet year with the PLHP (S3); 4: Dry year with the PLHP (S4). Our intention was to provide a scientific reference for the future water environment planning, water ecological management, and maintenance of the wetland.

## 2. Materials and Methods

### 2.1. Study Area

Poyang Lake is located in the middle of Yangtze River basin, Jiangxi Province, China (28°25′ N–29°45′ N, 115°48′ E–116°44′ E; Figure 1). It is a natural, seasonally impounding and releasing lake. The upper reaches are connected to the Ganjiang, Fuhe, Xinjiang, Rao, and Xiujiang Rivers. The lower reaches are connected to the Yangtze River (Figure 1). This lake is the one of the two lakes linked to the river in the Yangtze River system, and it is the largest freshwater lake in China. It is affected by the subtropical monsoon climate, with extreme differences in the water surface area and storage capacity between wet and dry seasons. In wet seasons, the mean surface area is 3572 km$^2$ and the mean storage is $280.5 \times 10^9$ m$^3$. In dry seasons, the mean surface area is 556.6 km$^2$ and the mean storage

is $9.2 \times 10^9$ m$^3$ [7]. The high, seasonal water level variation in Poyang Lake has developed a unique landscape with important ecological patterns and features (high species richness, large biomass, high biodiversity). Seven main hydrological stations exist in the main lake area of Poyang Lake. Among them, Xingzi Station is located at the bottleneck of the main river channel of Poyang Lake; therefore, its water level is often regarded by researchers as a representative of the overall water level of Poyang Lake [6]. In the wet season, Poyang Lake resembles a vast ocean. When the water level at the station is lower than 14.5 m, the dish-shaped lake starts to appear due to a depression in the basin during the dry season. The dish-shaped lake is a dish-shaped depression in the Poyang Lake Basin during the dry season. This lake provides a superior habitat for migratory birds due to its large area of wetland vegetation and zoobenthos developed in the spring. When the water level at the station drops to approximately 10 m, dish-shaped lakes are completely exposed and no longer connected to the main lake. This phenomenon marks the dry season in Poyang Lake [20]; at the same time, a large number of submerged plants and zoobenthos that developed during the wet season reach the migratory feeding conditions, and migratory birds come in succession. More than 300,000 birds visit Poyang Lake [6], including rare species such as *Grus leucogeranus*, *G. monacha*, *G. grus*, *G. vipio*, *Ciconia boyciana*, *Cygnus columbianus*, *Anser cygnoides*, and *A. albifrons*. There are three types of migratory bird nature reserves in the lake area, including national, provincial and county-level nature reserves.

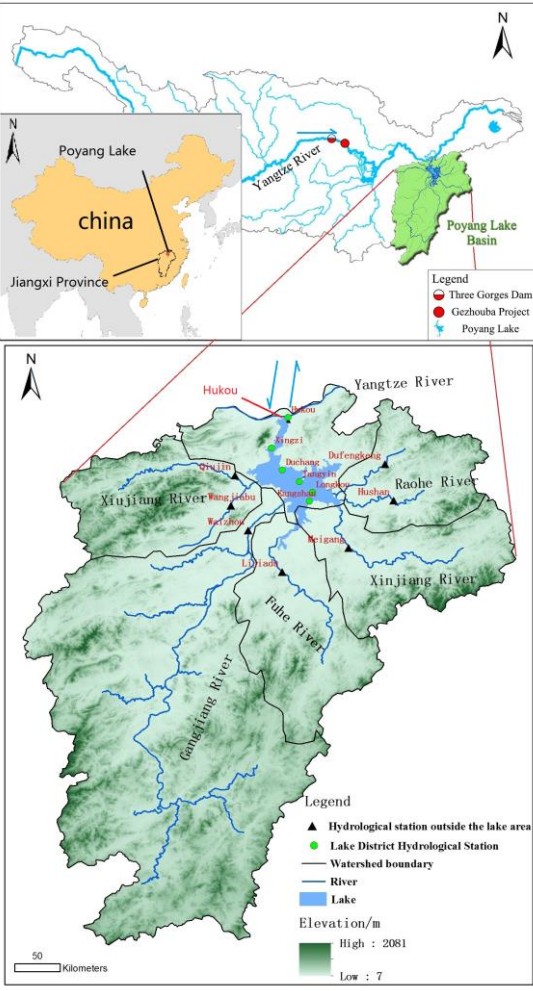

**Figure 1.** Study area. Upper image: Blue arrows indicate the flow direction in the Yangtze River. Lower image: Blue arrows indicate the flow direction in the water exchange channel between Poyang Lake and the Yangtze.

## 2.2. Approach for Habitat Suitability Evaluation

### 2.2.1. Selection of Evaluation Factors

The main environmental factors controlling the living conditions of the main migratory birds (*Grus leucogeranus*, *Grus monacha*, *Platalea leucorodia*, wintering Anseriformes, etc.) that winter in Poyang Lake were identified in accordance to previous studies [21–23] and the suggestions from the local conservation department of Poyang Lake. The factors selected for the migratory birds' habitat suitability evaluation were (1) the duration of floods in the wet season (represents the process of water level during the wet season), (2) landscape classification, (3) water depth, (4) vegetation coverage, (5) human activity distance, (6) the level of nature reserve. Wetland landscape variation results in varying food availability for migratory birds [6]. The water level process of Poyang Lake during the wet season influences the food abundance of the migratory birds [21]. Wintering cranes generally nest at a depth of 10–30 cm, and most of the migratory birds forage at a water depth of 20–60 cm [24,25]. Vegetation coverage provides rest places for wintering migratory birds and plays a vital role in their survival [26]. Migratory birds tend to stay away from human activities when they select habitats [27]. The Chinese government has divided two national, one provincial, and two county-level nature reserves in the Poyang Lake area, all of which play a key role in protecting the habitat of migratory birds in Poyang Lake. The aforementioned factors were divided into four categories, as shown in Table 1.

**Table 1.** Classification of the evaluation factors for habitat suitability for migratory birds.

| Target Level | Evaluation Level | Evaluation Standard | | | | Reference |
|---|---|---|---|---|---|---|
| | | $M_{01}$ Suitable Level | $M_{02}$ Less Suitable Level | $M_{03}$ Less Unsuitable Level | $M_{04}$ Unsuitable Level | |
| Habitat suitability for migratory birds | c1 duration of floods in the wet season | 90–123 days | 60–90 days | 30–60 days | 0–30 days | |
| | c2 landscape classification | Shallow water, quagmire | Sparse meadow, mire | Meadow, beach | Deep water | [6] |
| | c3 water depth | 0–0.6 m | 0.6–2 m | 2–10, 0 m | 10–Max m | [24,25,28] |
| | c4 vegetation coverage | 60%–100% | 45%–60% | 30%–45% | 0%–30% | [29] |
| | c5 human activity distance | 1500–Max m | 1000–1500 m | 500–1000 m | 0–500 m | [26] |
| | c6 level of nature reserve | National nature reserve | Provincial nature reserve | County-level nature reserve | Non-natural reserve | |

In this study, a comprehensive evaluation model for the habitat suitability for migratory birds was developed. The matter-element theory was used to comprehensively evaluate the habitat suitability on the geographic information system (GIS) software platform. The main steps were the following: (1) A factor evaluation system was established, and the analytic hierarchy process (AHP) was selected to determine factor weight; (2) a hydrodynamic model was established, and the hydrodynamic model, GIS [30], and remote sensing (RS) [31] were used to calculate the values of the various factors; (3) the matter-element analysis model was introduced into the ArcGIS platform to calculate single factor neartude; (4) on the ArcGIS platform, the factor comprehensive neartude was calculated, and the suitability of the composite factor was evaluated. A schematic representation of the steps followed in this study is shown in Figure 2.

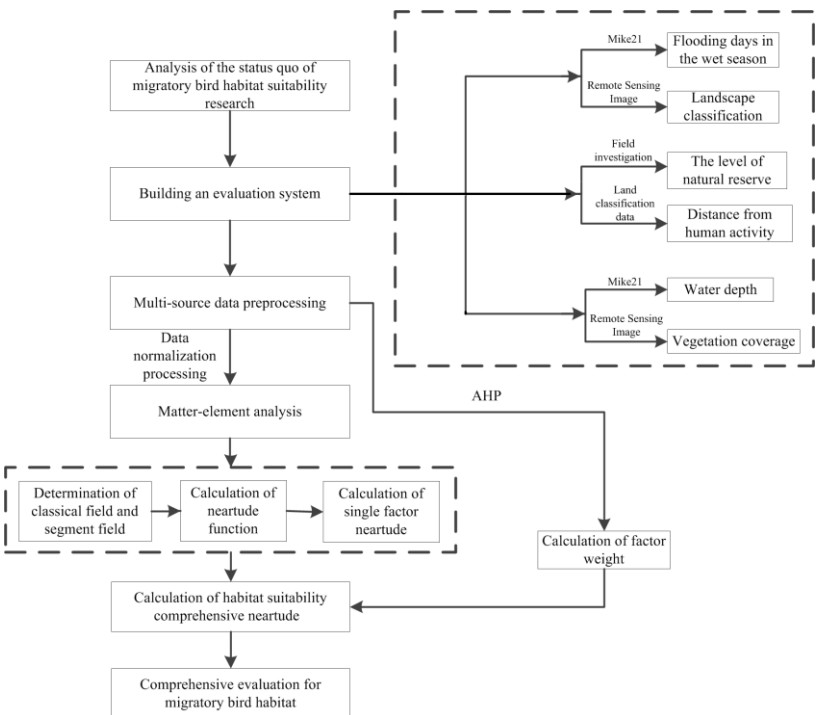

**Figure 2.** Schematic representation of the steps implemented to evaluate habitat suitability for migratory birds in Poyang Lake.

### 2.2.2. Selection of Typical Wet and Dry Years and Remote Sensing Images

Previous studies have shown that 2006 was a typical dry year according to the water level and the duration of dry season [32,33]. The average discharge in 2010 ranked second in the period 1983–2012 [34]. Thus, 2006 was selected as a typical dry year and 2010 as a typical wet year. Habitat suitability was evaluated for four scenarios: 1: Wet year without the PLHP (S1); 2: Dry year without the PLHP (S2); 3: Wet year with the PLHP (S3); 4: Dry year with the PLHP (S4).

The water level of Poyang Lake changes with the season, and it is controlled by the incoming water from the five rivers and the water level of the Yangtze River. Thus, the wintering time of migratory birds, the overall level of Poyang Lake, and the regulation water level of the PLHP were considered. When we selected the remote sensing image data, two images of the wet and dry years, which are the times when the water level of Xingzi Station and the regulation water level of the PLHP are similar, were considered. After comparison and analysis, the dates of remote images were selected as 6 June 2007 (the observed water level of the station was 6.00 m, the simulated water level was 6.18 m, and the regulation water level was 9.96 m) and 8 December 2010 (the observed water level of the station was 6.35 m, the simulated water level was 6.38 m, and the regulation water level was 10 m), which represent dry and wet years, respectively.

### 2.2.3. Calculation of the Habitat Suitability Evaluation Factors

#### 2D Hydrodynamic Model

(1) Establishment of a 2D Hydrodynamic Model

Poyang Lake has a complex shoreline and a highly variable terrain with considerable seasonal flow variability. In addition, it is a wide and shallow lake with vertical water mixing, and, thus, the flow field can be adequately simulated using a two-dimensional hydrodynamic model [35,36]. Based on the above, the Mike21 model (MIKE ZERO 2014, DHI, Denmark) [37], using an unstructured mesh, was selected to calculate the duration of floods in the wet season (c1) and water depth (c3).

1. Mesh partition: The topography of the lake was acquired from a digital elevation model developed in 2010. The shore boundary was defined based on the historical maximum water level recorded in the lake, as shown in Figure 3. Rivers and dish-shaped lakes within the model range were partially encrypted. The unstructured triangular mesh that was developed consisted of 206,970 elements and 106,475 nodes.

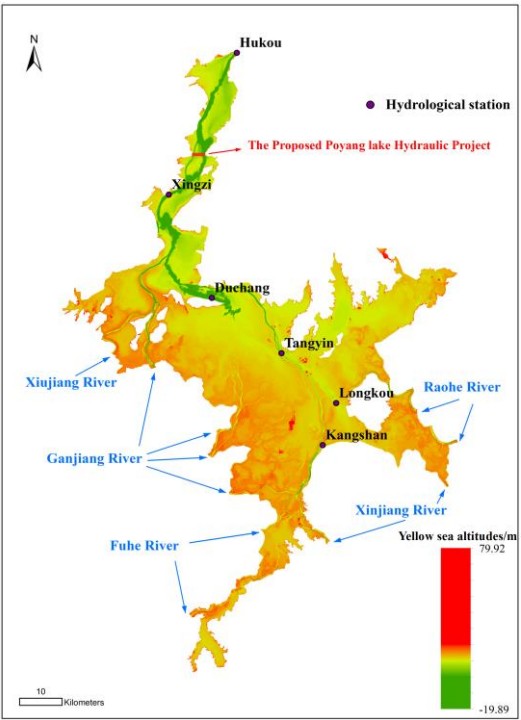

**Figure 3.** Topographical representation of Poyang Lake and its connected rivers. The blue arrows indicate the inflow open boundaries of the model. Hukou is the outflow open boundary. Altitude is the altitude of the Yellow Sea.

2. Boundary condition: We considered the discharge hydrograph of 11 main lake inlets of the 5 rivers (Ganjiang, Xiujiang, Fuhe, Rao, and Xinjiang) as the upstream open boundary conditions of the hydrodynamic model and the water level hydrograph in Hukou, which is called the water exchange channel of Poyang Lake and Yangtze River, as the downstream boundary condition. In addition, to account for inflows to Poyang Lake from unmonitored sources other than the eleven main inlets [22], the inflow data were multiplied by a coefficient (1.10) in order to maintain the water balance in the model [13] (achieving inflow approximately equal to the outflow in a simulated hydrological cycle).

3. Parameters determination: We used a spatially varying roughness coefficient based on topographical features, ranging from 0.018 in the river zone to 0.028 in the vegetation area of the beach.

4. Model validation: The model was verified using water level data from 2010–2011 for the Xingzi, Duchang, Kangshan, Longkou, Tangyin stations in the lake and using discharge data from 2010 for Hukou station. Due to the lack of the measured data of 2011 in Longkou Station and Hukou Station, we only verified the corresponding hydrograph of 2010. The water regimes in Poyang Lake for 2010–2011 were simulated by the established hydrodynamic model. The discharge and water level processes of the hydrological station (Figure 3) in the lake were extracted, and the observations were used to verify the simulation results. According to the Figure 4 and Table 2, the average water level error ranged from 0.09 to 0.40 m, the Nash–Sutcliffe (N–S) coefficient [38] (used to verify the quality of the hydrological and hydrodynamic model simulation results)

was between 0.909 and 0.990, and the N–S coefficient of Hukou Station was 0.926. The average discharge error was 741.79 m³/s. As such, the hydrodynamic model built in this study had good precision and could be used for later calculations.

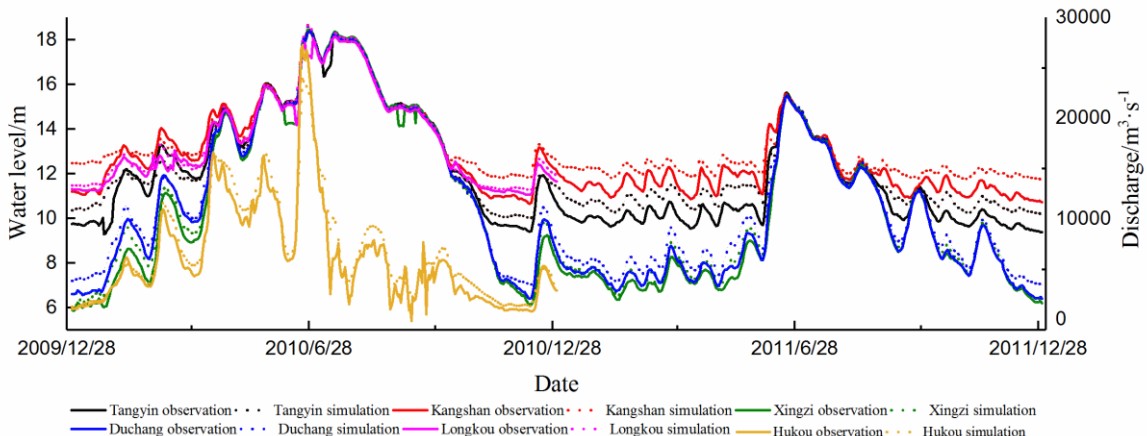

**Figure 4.** Verification of the hydrodynamic model. The yellow dots and line represent water discharge, and the rest represent the water level; the dotted line is the observation and the solid line is the simulation.

**Table 2.** Analysis of model error. The average water level error is the difference between the simulated average water level and the observed average water level.

| Station Name | Observed Water Level Average/m | Simulated Water Level Average/m | Average Water Level Error/m | Nash–Sutcliffe Coefficient |
|---|---|---|---|---|
| Xingzi | 11.9 | 12.1 | 0.2 | 0.99 |
| Tangyin | 13.32 | 13.43 | 0.09 | 0.95 |
| Longkou | 13.68 | 13.83 | 0.15 | 0.982 |
| Kangshan | 13.87 | 14.27 | 0.4 | 0.909 |
| Duchang | 12.25 | 12.45 | 0.2 | 0.985 |

(2)  Simulation of the Water Level Regulation of the PLHP

Migratory birds visit the lake from October to March (dry season). The PLHP plans to control the water level from September to March (Figure 5). The wet season starts from July and ends in October. Based on the above, our simulation period ranged from early May to early March.

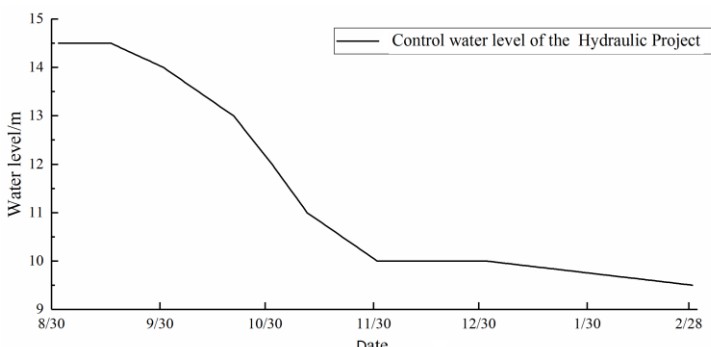

**Figure 5.** Water-level regulation process in the hydraulic project.

The structure setting of the Mike 21 flow model (FM) cannot meet the water-level-regulating requirements of the PLHP. Thus, an open boundary was defined at the location of the PLHP, and the water-level hydrograph of the PLHP was used in this boundary to simulate the regulative activity of the PLHP (Figure 5). The water level difference between the upper and lower gates in the actual

water storage process of the PLHP was considered, and the model was centered on the PLHP and divided into two submodels—the southern and the northern (Figure 3). The southern model had 196,532 elements with an area of 2846.1 km$^2$. The northern model had 10,438 elements with an area of 160.9 km$^2$. The south model simulated the flow field upstream of the PLHP, and the northern model simulated the flow field downstream of the PLHP. The inflow boundary condition for the southern model was the discharge hydrographs of the 11 main inlets of the five rivers; the outflow boundary condition for the southern model was the water level hydrograph of the PLHP. The discharge that was simulated for the outflow boundary of the southern model was used as the inflow boundary of the northern model, and the lower boundary of the northern model was the water level hydrograph of Hukou Station.

RS and GIS

The distance of human activity (c5) was calculated from the land classification data of Jiangxi Province in 2005 and 2010 by using the Euclidean distance measure (a tool for calculating the distance between two points) in the ArcGIS toolbox. The levels of nature conservation (c6) were directly from the data given by local departments in the field survey. c2 and c6 had no specific values, so the factors needed to be assigned. To quantify factors c2 and c6, the values of 0, 1, 3 and 5 were assigned to the unsuitable, less unsuitable, less suitable, suitable levels, respectively [39]. The source data of factors c2 (landscape classification) and c4 (vegetation coverage) were obtained by correcting and splicing LandSat images from 2007 (date: 6 June 2007) and 2010 (date: 8 December 2010) using ENVI software. After obtaining the source data, supervised classification was performed in ENVI to obtain c2 (landscape classification). A normalized difference vegetation index (NDVI) was used to estimate c4 (vegetation coverage). The NDVI can not only partly eliminate the influences of terrain, atmosphere, and shadow, it can also effectively distinguish vegetation from rock and bare soil, which is the best indicator of vegetation coverage [29]. The NDVI was calculated in ENVI software based on the LandSat images. The following equation [29] was used to calculate vegetation coverage in ENVI software:

$$F = \frac{NDVI - NDVI_{soil}}{NDVI_{veg} - NDVI_{soil}} \tag{1}$$

where *F* represents the vegetation coverage of the habitat studied area, $NDVI_{soil}$ represents the NDVI with a cumulative frequency of 5%, and $NDVI_{veg}$ represents the NDVI with cumulative frequency of 95%. Cumulative frequency is the percentage of the frequency of the NDVI value that accounts for the total frequency in the area. The NDVI with cumulative frequency could be found in ENVI software after NDVI was calculated.

Correction of Evaluation Factors

The remote sensing images with the PLHP were needed to calculate the landscape classification factor c2 and vegetation coverage factor c4 in S3 and S4, but such remote sensing images do not exist at present. In this study, c2 and c4 factors in S3 and S4 were obtained by the following way:

1.  For the southern region (from the entrance of the five rivers to the PLHP (Figure 3): The corresponding hydrodynamic model simulation results of scenes with the PLHP were used to correct the landscape classification results and vegetation cover results of remote sensing images without the PLHP. For the results of landscape classification, the hydrodynamic model was used to simulate water depth to correct the deep and shallow water areas in the remote sensing images, but the rest of the area was unchanged. For the results of vegetation coverage, the vegetation coverage on the water surface was assigned a value of 0 by the water surface simulated by the hydrodynamic model, but the rest of the area was unchanged.

2.  For the northern region (from the location of the PLHP to Hukou Station (Figure 3): In the hydrodynamic model, the PLHP downstream water level processes of S1 and S2 were compared

with those of S3 and S4, respectively, and the difference in water level process was minimal. Therefore, the landscape classification and vegetation coverage results of S1 and S2 were used as the results of S3 and S4 of those factors, respectively, and they were included in the calculation.

### 2.2.4. Normalization of the Habitat Suitability Evaluation Factors

After the original value of each factor was obtained, the following equations were used for normalization to eliminate the difference in dimension.

For the positive factor, the following equation was used:

$$A_{ij} = \frac{X_{ij} - minX_{ij}}{maxX_{ij} - minX_{ij}} \tag{2}$$

For the negative factor, the following equation was adopted:

$$A_{ij} = \frac{maxX_{ij} - X_{ij}}{maxX_{ij} - minX_{ij}} \tag{3}$$

where $A_{ij}$ is the normalized value, $X_{ij}$ is the original value of the *j*-th factor in the *i*-th year, and $minX_{ij}$ and $maxX_{ij}$ are the minimum and maximum values of the *j*-th factor, respectively.

### 2.3. Matter Element Model

Matter element theory: Matter-element theory is a new subject to study the regularity problems under the comprehensive action of multiple factors [40,41]. The matter-element model can be used to solve incompatible complex problems, and it is suitable for multifactor evaluation [40,42,43]. Thus, the matter-element evaluation method has been widely applied in the quantitative comprehensive evaluation of land ecological security, water ecological civilization, and water quality [43–50]. In this study, the matter-element theory was used to construct a habitat suitability evaluation model for migratory birds in Poyang Lake.

The main steps of building this matter-element model are as follows:

1.  Building of Matter Element:

In the matter-element analysis, the described object (the habitat suitability for migratory birds) *M*, its eigenvector *c*, and the eigenvalue *v* constitute the habitat suitability for migratory birds, $R = (M, c, v)$, where *M*, *c*, and *v* are the three elements of matter element *R*. If the object *M* has n eigenvectors $c_1$, $c_2$, ... , $c_n$ and its corresponding eigenvalues $v_1$, $v_2$, ..., $v_n$, then *R* is the suitability matter-element of n-dimensional migratory bird habitat, and the corresponding matter-element matrix can be expressed as:

$$R = (M, c, v) = \begin{bmatrix} M & c_1 & v_1 \\ & c_2 & v_2 \\ & \vdots & \vdots \\ & c_n & v_n \end{bmatrix} = \begin{bmatrix} \textit{Habitat suitability for migratory birds} & \textit{Flood duration} & v_1 \\ & \textit{Landscape classification} & v_2 \\ & \textit{Water depth} & v_3 \\ & \textit{Vegetation coverage} & v_4 \\ & \textit{Distance of human activity} & v_5 \\ & \textit{Level of nature protection} & v_6 \end{bmatrix} \tag{4}$$

2.  Determination of the Classical and Segment Fields:

The classical field matter-element matrix of habitat suitability for migratory birds can be expressed as $R_{oj} = (M_{oj}, c_i, v_o)$, where $R_{oj}$ is the classical field matter-element of habitat suitability for migratory birds, $M_{oj}$ is the *j*-th evaluation level of habitat suitability for migratory birds ($j$ = 1, 2, ... , n), $c_i$ represents the *i*-th evaluation factor, and $(a_{oj1}, b_{oj1})$ is the range of the eigenvectors $c_i$ corresponding to

the evaluation level $M_{oj}$. Therefore, the classical field matter-element matrix of habitat suitability for migratory birds can be expressed as:

$$R_{oj} = \left(M_{oj}, c_i, v_o\right) = \begin{bmatrix} M_{oj} & c_1 & \left(a_{oj1}, b_{oj1}\right) \\ & c_2 & \left(a_{oj2}, b_{oj2}\right) \\ & \vdots & \vdots \\ & c_n & \left(a_{ojn}, b_{ojn}\right) \end{bmatrix} \tag{5}$$

The classical field $(R_{01}, R_{02}, R_{03}, R_{04})$ is expressed as:

$$R_{01} = \begin{bmatrix} M_{01} & c_{01} & (0.73, 1) \\ & c_{02} & (0.60, 1) \\ & \vdots & \vdots \\ & c_{06} & (0.60, 1) \end{bmatrix} R_{02} = \begin{bmatrix} M_{02} & c_{01} & (0.49, 0.73) \\ & c_{02} & (0.40, 0.60) \\ & \vdots & \vdots \\ & c_{06} & (0.40, 0.60) \end{bmatrix}$$

$$R_{03} = \begin{bmatrix} M_{03} & c_{01} & (0.24, 0.49) \\ & c_{02} & (0.20, 0.40) \\ & \vdots & \vdots \\ & c_{06} & (0.20, 0.40) \end{bmatrix} R_{04} = \begin{bmatrix} M_{04} & c_{01} & (0, 0.24) \\ & c_{02} & (0, 0.20) \\ & \vdots & \vdots \\ & c_{06} & (0, 0.20) \end{bmatrix}$$

Similarly, the segment field matrix of habitat suitability for migratory birds can be expressed as:

$$R_p = (M_p, c_n, v_p) = \begin{bmatrix} M_p & c_1 & (a_{oj1}, b_{oj1}) \\ & c_2 & (a_{oj2}, b_{oj2}) \\ & \vdots & \vdots \\ & c_6 & (a_{ojn}, b_{ojn}) \end{bmatrix} = \begin{bmatrix} P & c_{01} & (0, 1) \\ & c_{02} & (0, 1) \\ & \vdots & \vdots \\ & c_{06} & (0, 1) \end{bmatrix} \tag{6}$$

where $R_p$ is the segment field matter-element, $p$ is the habitat suitability level of migratory birds, and $v_p$ is the range of segment field matter-element on the characteristic $c_i$.

3.　Determination of Single Factor Neartude of Each Evaluation Level

$$K_j(v_i) = \begin{cases} -\dfrac{\rho(v_i, v_{oji})}{|v_{oji}|}, & (v_i \in v_{oji}) \\ \dfrac{\rho(v_i, v_{oji})}{\rho(v_i, v_{pi}) - \rho(v_i, v_{oji})}, & (v_i \notin v_{oji}) \end{cases} \tag{7}$$

where,

$$\begin{cases} \rho(v_i, v_{oji}) = \left|v_i - \frac{1}{2}(a_{oji} + b_{oji})\right| - \frac{1}{2}(b_{oji} - a_{oji}) \\ \rho(v_i, v_{pi}) = \left|v_i - \frac{1}{2}(a_{oji} + b_{oji})\right| - \frac{1}{2}(b_{pi} - a_{pi}) \end{cases} \tag{8}$$

where the value of the neartude $K$ on the real number axis represents the degree to which the object to be evaluated belongs to a certain standard of habitat suitability for migratory birds; $K_j(v_i)$ is the neartude of the $i$-th factor corresponding to the $j$-th level; and $\rho(v_i, v_{oji})$ and $\rho(v_i, v_{pi})$ represent the distance between point $v_i$ and the classical fields $v_{oji}$ and $v_{pi}$, respectively. $v_i$, $v_{oji}$, and $v_{pi}$ represent the value of the matter element to be evaluated, the value range of the classical field, and the value range of the segment field, respectively.

4.　Determination of the Weight of Each Factor

In this paper, the AHP was used to calculate the weight of each factor, and the weights were ranked according to the evaluation factors c1–c6: $\overline{W} = [0.127\ 0.315\ 0.091\ 0.201\ 0.133\ 0.133]$, and $CR = {}^{CI}/_{RI} = 0.065 < 0.1$. Thus, it met the consistency requirements.

5.　Calculation of Comprehensive Neartude and Determination of Evaluation Level

The comprehensive neartude of the object $Mx$ ($x = 1, 2, \ldots, m$) to be evaluated on level $j$ is

$$K_j(M_x) = \sum_{i=1}^{m} w_i K_j(v_i) \tag{9}$$

where $K_j(v_i)$ is the single factor neartude of $M_x$ on level $j$, and $w_i$ is the weight of factor $i$.

If

$$K_{jx} = \max K_j(M_x), \tag{10}$$

then the object $M_x$ belongs to the standard level $j$ of suitability of migratory bird habitat.

## 3. Results

### 3.1. Single Factor Evaluation of Habitat Suitability for Migratory Birds

The suitability of factor c1 was similar in wet years, with or without the PLHP (S1 and S3 scenarios) (Figures 6a and 7a), but it was significantly different in dry years, in which the proportion of suitable habitats increased by 40.1% when the PLHP was operating (S4 scenario, in contrast to S2) (Figures 6a and 7b). Factor c2 exhibited the highest variation between the four scenarios (Figure 6b). The percentage of unsuitable habitats increased when the PLHP was operating, both in wet and dry years (18.77% increase in S3 compared to S1 (Figure 7a) and 16.66% increase in S4 compared to S2 (Figure 7b)). Factor c3 did not show any variation between wet and dry years, with or without the operation of the PLHP (Figures 6c and 7). Suitable and less suitable habitats slightly increased when the PLHP was operating in both wet and dry years (S3 and S4 compared with S1 and S2). Regarding factor c4 (Figure 6d), the percentage of unsuitable habitats increased in both project-operating scenarios (S3 and S4) due to the expansion of the water surface (29.57% increase between S3 and S1 (Figure 7a); 25.05% increase between S4 and S2 (Figure 7b)). Factors c5 and c6 were not assessed, as they were independent of the operation of the PLHP.

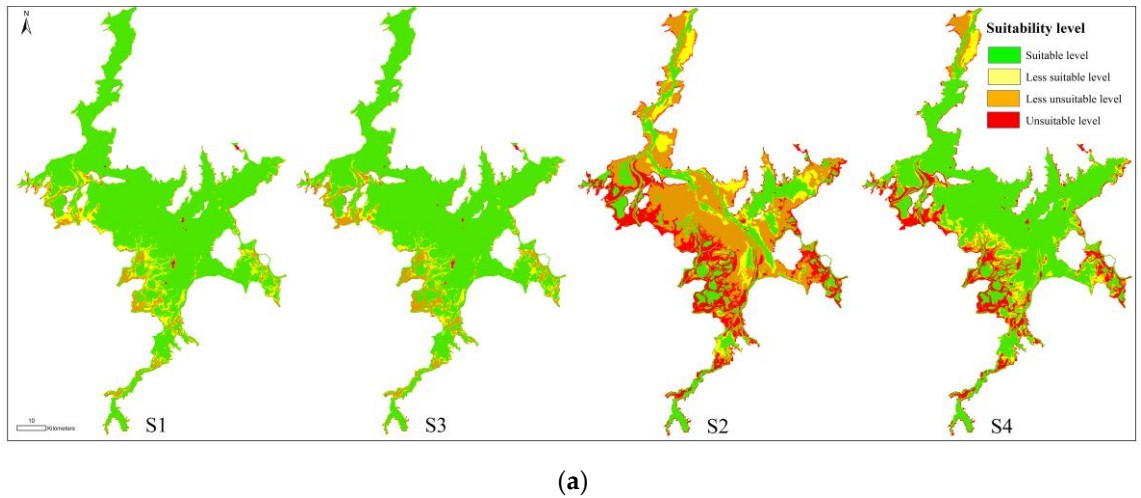

(**a**)

**Figure 6.** *Cont.*

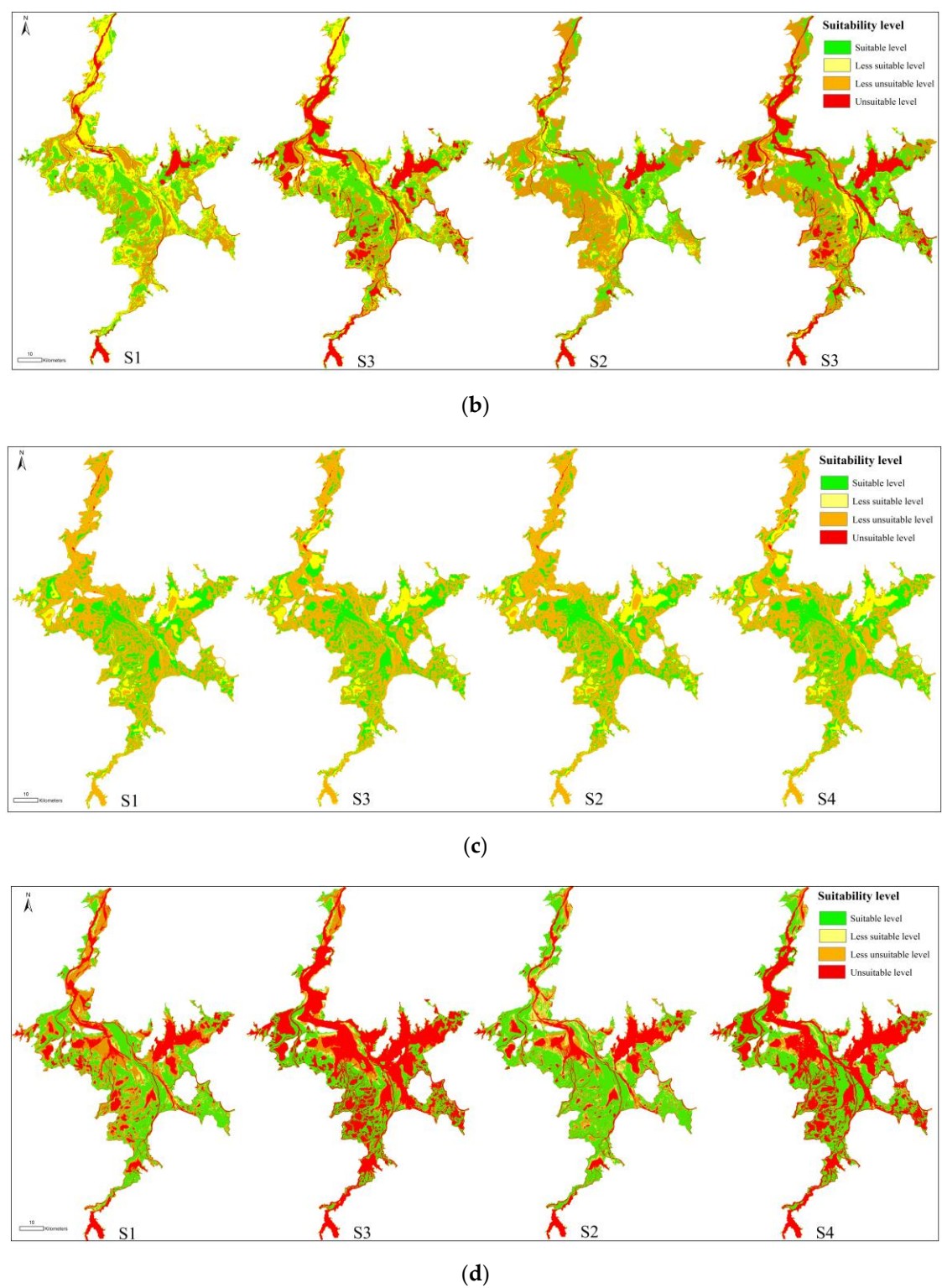

**Figure 6.** Migratory-bird habitat suitability of each selected habitat evaluation factor in Poyang Lake at each studied scenario. c1: Flood duration; c2: Landscape classification; c3: Water depth; c4: Vegetation coverage; c5: Distance of human activity; c6: Level of nature protection. S1: Wet year, no hydraulic project; S2: Dry year, no hydraulic project; S3: Wet year, hydraulic project present/operating; S4: Dry year, hydraulic project present/operating. (**a**) Single factor suitability of c1; (**b**) single factor suitability of c2; (**c**) single factor suitability of c3; (**d**) single factor suitability of c4.

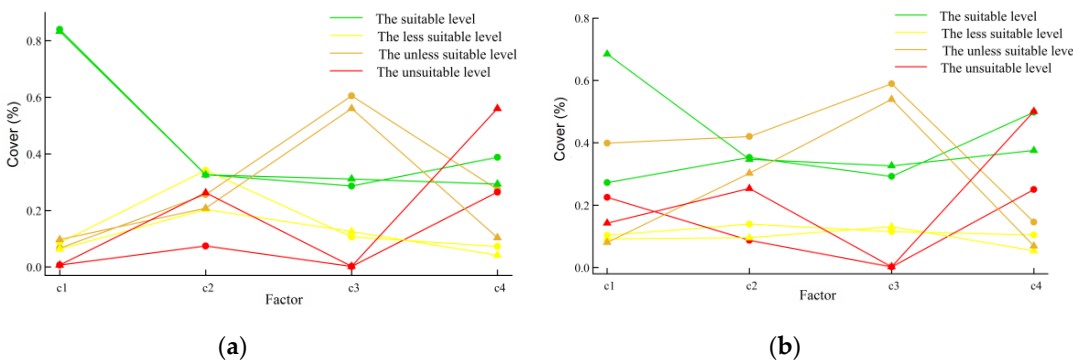

**Figure 7.** Percent habitat-level change per factor in Poyang Lake in the four scenarios; ● represents S1 in (**a**) and S2 in (**b**), and ▲ represents S3 in (**a**) and S4 in (**b**). S1: Wet year, no hydraulic project; S2: Dry year, no hydraulic project; S3: Wet year, hydraulic project present/operating; S4: Dry year, hydraulic project present/operating. (**a**) Percent habitat-level change between S1 and S3; (**b**) percent habitat-level change between S2 and S4.

### 3.2. Comprehensive Evaluation of Habitat Suitability for Migratory Birds

In wet years, the comprehensive habitat suitability varied based on the operation of the hydraulic project or not (S1 and S3 scenarios) (Figure 8a,b). The less unsuitable and unsuitable habitats increased when the hydraulic project was operating. These habitats were mainly distributed in and around the proposed PLHP location to the southern provincial nature reserve and some dish-shaped lakes. In dry years, the comprehensive habitat suitability also depended on the operation of the hydraulic project (S2 and S4 scenarios) (Figure 8c,d). Suitable, less suitable and unsuitable habitats increased when the hydraulic project was operating. Suitable habitats increased in and around the southern national nature reserve. The change of less unsuitable and unsuitable habitats in dry year were similar to that in the wet year.

In wet years, the percentage of suitable habitats and less suitable habitats decreased when the hydraulic project was operating (17.6% decrease in S3 compared to S1 (Figure 9 and Table 3)). Thus, the operation of the hydraulic project increases the water depths in parts of the lake, making these habitats unsuitable for perching, on which the migratory birds feed. On the contrary, in dry years, the percentage of suitable habitats and less suitable habitats increased when the hydraulic project was operating in dry year (4.08% increase in S3 compared to S1 (Figure 9 and Table 3)). Though the operation of the proposed construction of the PLHP increases the water depth in parts of the habitat, it obviously increases the submerged duration of the lake in dry years, alleviating the ecological water demand of the wetland, and increasing the food abundance of migratory birds in the wetland.

**Table 3.** Area (km²) of habitat suitability levels and the proportion of each level to the total area of Poyang Lake at each of the four scenarios. S1: Wet year, no hydraulic project; S2: Dry year, no hydraulic project; S3: Wet year, hydraulic project present/operating; S4: Dry year, hydraulic project present/operating.

| Suitability Level | S1 | | S3 | | S2 | | S4 | |
|---|---|---|---|---|---|---|---|---|
| | Area | Proportion | Area | Proportion | Area | Proportion | Area | Proportion |
| Suitable level | 1456.30 | 48.43% | 1184.10 | 39.37% | 922.5 | 30.67% | 1175.30 | 39.12% |
| Less suitable level | 983.4 | 32.70% | 726.5 | 24.16% | 553 | 18.39% | 421.4 | 14.03% |
| Less unsuitable level | 411.8 | 13.69% | 718.3 | 23.88% | 1324.10 | 44.03% | 1004.20 | 33.42% |
| Unsuitable level | 155.6 | 5.17% | 378.7 | 12.59% | 207.7 | 6.91% | 403.6 | 13.43% |

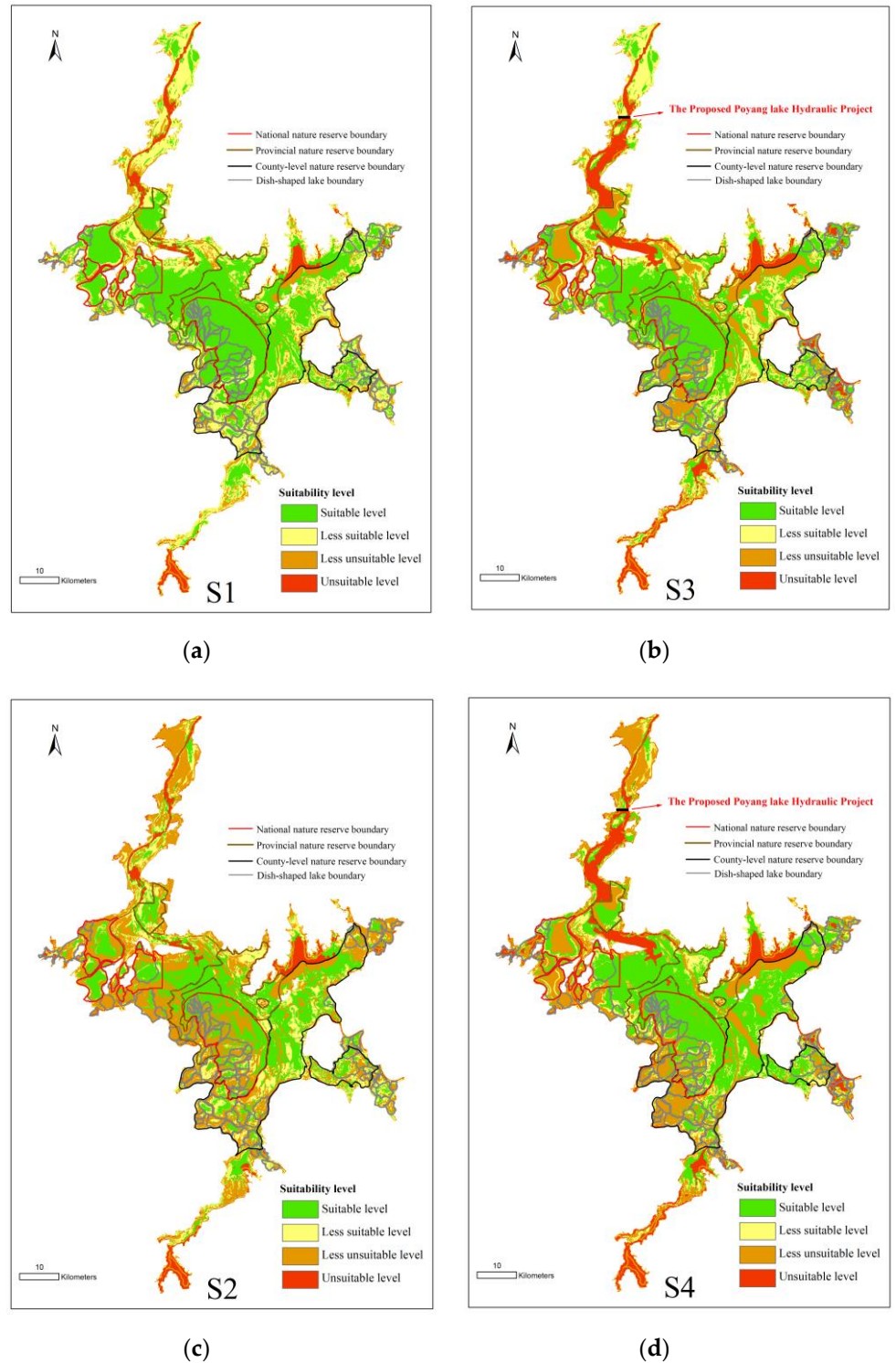

**Figure 8.** Results of the comprehensive evaluation of habitat suitability for migratory birds. (S1): Wet year, no hydraulic project; (S2): Dry year, no hydraulic project; (S3): Wet year, hydraulic project present/operating; (S4): Dry year, hydraulic project present/operating. (**a**) Suitability distribution for S1; (**b**) suitability distribution for S3; (**c**) suitability distribution for S2; (**d**) suitability distribution for S4.

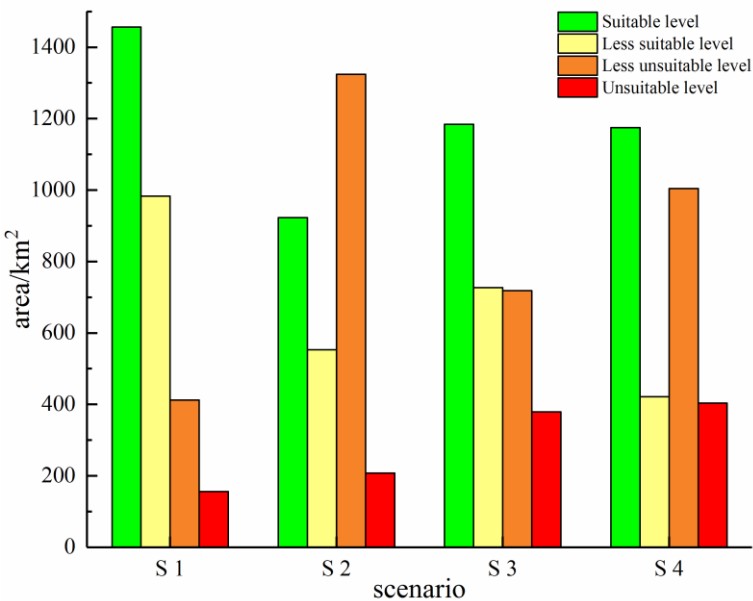

**Figure 9.** Area distribution of habitat suitability level. (S1): Wet year, no hydraulic project; (S2): Dry year, no hydraulic project; (S3): Wet year, hydraulic project present/operating; (S4): Dry year, hydraulic project present/operating.

### 3.3. Impact of the PLHP on Migratory-Bird Habitats in Each Nature Reserves

Most of the migratory birds in Poyang Lake are concentrated in all levels of nature reserves [11], so the nature reserves in the lake were further analyzed. In wet year, suitable and less suitable habitats decreased when the PLHP was operating in all levels of nature reserves (Table 4). Provincial nature reserves had the greatest change in the habitats. On the contrary, in dry years, suitable and less suitable habitats increased when the PLHP was operating in most nature reserves (Table 4). National nature reserves had the greatest change in the habitats. Therefore, in the wet year, the impact of the PLHP on the suitability for migratory birds was seen as mostly negative. In dry years, the impact was seen as mostly positive.

**Table 4.** Area (km$^2$) and percent change of habitat suitability levels between the studied scenarios at each nature reserve. N, P and C denote the national, provincial and county nature reserves. Percentages were calculated by dividing the area of each reserve by the total area of all nature reserves. Percent change was calculated as "percentage without the hydraulic project" minus "percentage with the hydraulic project". + and − denote the positive and negative impact, respectively, of the hydraulic project on habitat suitability.

| Suitability Level | | S1 | | S3 | | Change in the Proportion | Effect |
|---|---|---|---|---|---|---|---|
| | | Area | Proportion | Area | Proportion | | |
| Suitable level | N | 489.3 | 86% | 386.7 | 69.71% | 16.29% | − |
| | P | 231.2 | 74% | 166.9 | 53.68% | 20.32% | − |
| | C | 350 | 45% | 276.2 | 35.32% | 9.68% | − |
| Less suitable level | N | 62.8 | 11.33% | 57.7 | 10.40% | 0.93% | − |
| | P | 58.2 | 18.71% | 57.2 | 18.40% | 0.31% | − |
| | C | 277.2 | 35.44% | 222.5 | 28.45% | 6.99% | − |
| Less unsuitable level | N | 13.4 | 2.41% | 110.1 | 19.85% | −17.44% | − |
| | P | 21.6 | 6.94% | 86.6 | 27.85% | −20.91% | − |
| | C | 151.4 | 19.36% | 272.7 | 34.87% | −15.51% | − |
| Unsuitable level | N | 0 | 0 | 0 | 0 | 0 | 0 |
| | P | 0 | 0 | 0 | 0 | 0 | 0 |
| | C | 3.6 | 0.46% | 10.7 | 1.37% | −0.91% | − |

**Table 4.** *Cont.*

| Suitability Level | | S2 | | S4 | | Change in the Proportion | Effect |
|---|---|---|---|---|---|---|---|
| | | Area | Proportion | Area | Proportion | | |
| Suitable level | N | 241.3 | 43.51% | 316.9 | 57.15% | −13.64% | + |
| | P | 135.6 | 43.60% | 166.4 | 53.52% | −9.92% | + |
| | C | 284.3 | 36.65% | 355 | 45.39% | −8.74% | + |
| Less suitable level | N | 34.3 | 6.18% | 57.1 | 10.30% | −4.12% | + |
| | P | 56.8 | 18.26% | 54.5 | 17.53% | 0.73% | − |
| | C | 186.9 | 23.89% | 99.3 | 12.70% | 11.19% | − |
| Less unsuitable level | N | 278.2 | 50.16% | 179.5 | 32.37% | 17.79% | + |
| | P | 118.4 | 38.07% | 89.5 | 28.79% | 9.28% | + |
| | C | 304.5 | 38.93% | 314.2 | 40.17% | −1.12% | − |
| Unsuitable level | N | 1 | 0.14% | 1 | 0.14% | 0 | 0 |
| | P | 0 | 0 | 1 | 0.14% | -0.14% | − |
| | C | 6.5 | 0.83% | 13.6 | 1.74% | -0.91 | − |

## 4. Discussion

### 4.1. The Local Fishing Way, Namely "Lake Enclosed in Autumn"

The results showed that in some of the dish-shaped lakes of Poyang Lake (about 30% in a wet year, 15% in a dry year), the migratory bird habitat suitability was higher without the presence of the hydraulic project and decreased with the operation of the hydraulic project. This outcome, however, should be interpreted in combination with a factor that cannot be hydrodynamically simulated: A unique, traditional fish production habit, the "lake enclosed in autumn" [51]. During autumn and winter, when the dish-shaped lake is formed, local fishermen control the water level by establishing a drainage gate on the drain in the dish-shaped lake to fish in the outlet. Thus, Poyang Lake becomes semi-naturally controlled. This process cannot be simulated. Consequently, the simulated surface area and water level of the lake during the dry season could be different from the actual one. If the lake is not controlled by the "lake enclosed in autumn" habit, most of the dish-shaped lake's water depth will be 1 m in the dry season, and this is not suitable for the migratory birds to inhabit and feed (Figure 10). The "lake enclosed in autumn" can lower the water level to meet suitable habitat conditions for migratory birds. Yet, the unreasonable release of water from the dish-shaped lake will probably cause aquatic plants and zoobenthos in the dish-shaped lake to die prematurely and thus reduce the food available for migratory birds. The key to ensure suitable habitats for the migratory birds is to release water from the dish-shaped lake only when it is necessary, considering the aforementioned process. Our results showed that a scientific and rational management of the water releases from the dish-shape lake can benefit both the fishermen's habit and the migratory bird habitats.

This habit has a general regularity [51] (the fishermen close and open the gates in about July and November, respectively). In addition, it depends on the overall water level of Poyang Lake (the earlier low water level makes the closing gate early). Therefore, in order to further evaluate Poyang Lake, the regularity was studied based on high-frequency and long-term remote sensing images.

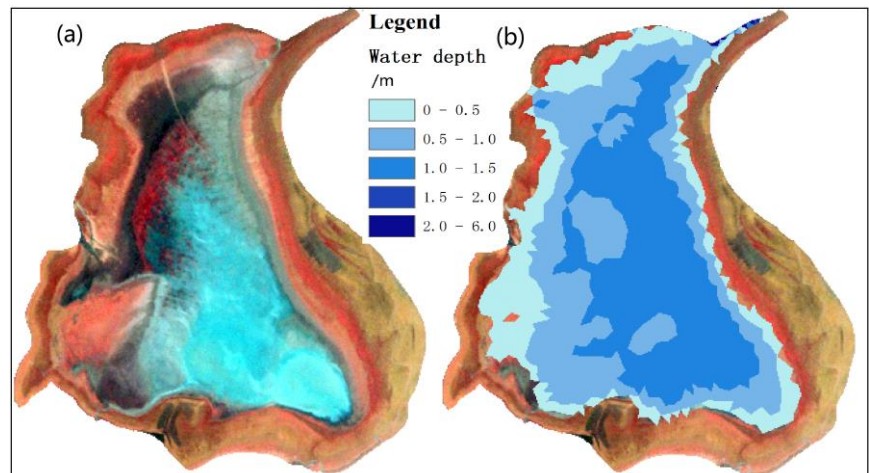

**Figure 10.** Comparison of actual and simulated water levels in a dish-shaped lake (Bang Lake). (**a**) represents the actual water surface, (**b**) represents the simulated water surface. The legend represents the depth of (**b**), and the water in (**a**) is blue.

### 4.2. The Increased Unsuitable Area in S3, S4

Based on the results, after the PLPH operation, there will be an increased number of unsuitable habitats (5.17% in S1, 12.59% in S3; 6.91% in S2, 13.43% in S4 (Figure 9). These habitats were mainly distributed in the PLHP location to the southern provincial nature reserve (Figure 8). This is mainly because the PLHP operation raises the water level by about 3 m (Figure 5) and causes the original habitat to become deep water, which reduces the suitability. However, migratory birds are not concentrated in these areas [11].

### 4.3. Correction and Comparison of the Evaluation Results

The process of regulating the water level within the proposed construction of the PLHP causes the water in the lake to drop to approximately 10 m in December. At this time, the dish-shaped lake becomes disconnected from the main Poyang Lake, and, thus, the dish-shaped lake is not affected by the operation of the PLHP. In and around the national nature reserve, the difference in the habitat suitability with and without the presence/operation of the PLHP was observed mainly in the dish-shaped lake. Based on the proportion of the suitable and less suitable areas, the provincial and county-level nature reserves and their surrounding areas connected to the main lake will be more negatively affected (Table 4). The provincial nature reserves in the north will be most impacted (30.32% decrease in S3 compared to S1; 16.43% decrease in S4 compared to S2).

According to Wang et al. [12], the project will have little impact on the national nature reserves. In contrast, our study is in partial accordance with Guo et al. [11], who concluded that the project will have a great impact on the provincial nature reserves; it will lose half of the area suitable for migratory birds. Worldwide, Niekerk et al. [52] noted that increasing the area of shallows at the inflow of larger dams (Highveld, South Africa) in open landscapes with fringing vegetation may encourage geese (*Plectropterus gambensis*) breeding. Bejarano et al. [53] noted that changes in dispersal patterns due to hydropower infrastructure may cause aquatic vegetation species distributions to shift towards the upper boundaries of regulated rivers. Algarte et al. [54] noted that changes in hydrometric levels caused by dam operations can alter community structure in aquatic environments and short-term emersion affected the structure and composition of the periphyton community in the Upper Parana' floodplain (Brazil). Currently, Poyang Lake has a long-term water shortage during the dry season, and the operation of the PLHP will effectively solve this problem, but our results showed that areas suitable for migratory birds will be lost.

### 5. Conclusions and Suggestions

The study of the impact of the PLHP on the wintering migratory bird habitats is the key factor that determines the implementation of the project or not. In this study, we developed a model to evaluate and simulate the habitat suitability for migratory birds in Poyang Lake before and after the establishment and operation of the PLHP. The impact of the operation of the PLHP on the habitat suitability distribution of migratory birds was predicted.

The following conclusions were drawn: The PLHP can effectively alleviate the water shortage problem in the wetland and increase the proportion of suitable migratory bird habitats in typical dry years. However, it will cause a 30.32% reduction of the suitable habitats in the northern provincial nature reserve. Therefore, the PLHP should develop dynamic dispatching rules. For example, the lake's water level should drop when migratory birds arrive to increase habitat availability. In addition, a scientific management of the "lake enclosed in autumn" habit by the local government can increase the migratory bird habitat availability-suitability and promote harmonic relationships between people and migratory birds. A dish-shaped lake with a water depth of more than 1 m can be drained. The time to release water needs to be prolonged to early February. This will ensure extended foraging space and habitat area for migratory birds, thus alleviating the risk of diseases and ecological stress caused by migratory birds overcrowding. This study provides a scientific reference for the future water environment planning, water ecological management, and maintenance of the Poyang Lake area.

The operation of the PLHP will increase the flooding of Poyang Lake. The vegetation community and biodiversity of the lake will be affected. The *Cynodon dactylon* community and the *Oncomelania hupensis Gredler* (the dominant plant and animal species of Poyang Lake) will be greatly damaged because of their extremely poor Submerged adaptability [19,55]. The PLHP may prevent the finless porpoise from returning and cause a decrease in water quantity in the Yangtze River. These impacts may become the focus of research in the future. The PLHP should explore a suitable ecological regulation water level as soon as possible, so that the lake area can reach the demand of the wetland dynamic characteristics to the water level. Based on the results, the northern provincial nature reserve will be greatly affected; thus, the nature reserve can be adjusted to other places in future management and planning of Poyang Lake.

**Author Contributions:** Conceptualization, S.Y. and C.L.; methodology, S.Y. and C.L.; software, X.L. and W.K.; validation, S.Y. and X.L.; formal analysis, S.Y. and X.L.; investigation, S.Y.; resources, C.L.; data curation, S.Y. and C.L.; writing—original draft preparation, S.Y. and L.Z.; writing—review and editing, D.Y. and X.M.; visualization, J.Y. and G.W.; supervision, C.L.; project administration, S.Y. and C.L.; funding acquisition, C.L.

**Funding:** This work is supported by National Natural Science Foundation of China under the contract No. 51709142 and No. 41261053; Jiangxi Water Conservancy Science and Technology Plan Project No. KT201326 and No. KT201616; Jiangxi Key Science and Technology Project No. 20171BBG70059; Jiangxi Province Graduate Innovation Special Fund Project (YC2018-S121).

**Conflicts of Interest:** The authors declare no conflict of interest. The funders had no role in the design of the study; in the collection, analyses, or interpretation of data; in the writing of the manuscript, or in the decision to publish the results.

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
