# Peer review of "Quantitative Assessment of Impact of the Proposed Poyang Lake Hydraulic Project (China) on the Habitat Suitability of Migratory Birds"

_water, doi:10.3390/w11081639_

Round 1

Reviewer 1 Report

I enjoyed reading your paper, found it well written and scientifically sound. I have only one minor comment.  The sentence beginning on line 68 sounds definitive when I don't think you intend that. I suggest inserting the word currently to clarify that you are talking about the present. The sentence would then read:

"However, it is difficult to establish a comprehensive habitat evaluation model after the completion of the PLHP, because the water regime (discharge, water level, flow velocity, water surface, etc.) is currently unknow.

Author Response

Dear Editors and Reviewer 1:

Thank you for your letter and for the reviewers’ comments concerning our manuscript entitled “Quantitative assessment of the impact of the proposed Poyang Lake Hydraulic Project (China) on the habitat suitability of migratory birds”. Those comments are all valuable and very helpful for revising and improving our paper, as well as the important guiding significance to our researches. We have studied comments carefully and have made correction which we hope meet with approval. Revised portion are marked in red in the paper. The main corrections in the paper and the responds to the reviewer’s comments are as flowing:

 The sentence beginning on line 68 sounds definitive when I don't think you intend that. I suggest inserting the word currently to clarify that you are talking about the present. The sentence would then read:

"However, it is difficult to establish a comprehensive habitat evaluation model after the completion of the PLHP, because the water regime (discharge, water level, flow velocity, water surface, etc.) is currently unknow.

Response: According to your suggestion, it has been revised.

Reviewer 2 Report

A brief explanation would be useful, what is the Nash-Sutcliffe (NS) coefficient (line 193), and why was it used (perhaps with a reference for it).

It is understandable that the „Lake Enclosed in Autumn” fishing way can not be hydrodinamically simulated. But it would be interesting if the authors would be able to present (also int he discussion section) some future research directions, which can contribute to impact assessement of that fishing habit.

Some spelling mistakes can be found in the text, a strict re-reading and corrections are needed.

Author Response

Dear Editors and Reviewer 2:

Thank you for your letter and for the reviewers’ comments concerning our manuscript entitled “Quantitative assessment of the impact of the proposed Poyang Lake Hydraulic Project (China) on the habitat suitability of migratory birds”. Those comments are all valuable and very helpful for revising and improving our paper, as well as the important guiding significance to our researches. We have studied comments carefully and have made correction which we hope meet with approval. Revised portion are marked in red in the paper. The main corrections in the paper and the responds to the reviewer’s comments are as flowing:

A brief explanation would be useful, what is the Nash-Sutcliffe (NS) coefficient (line 193), and why was it used (perhaps with a reference for it).

Response: A explanation and reference have been added.

It is understandable that the „Lake Enclosed in Autumn” fishing way can not be hydrodinamically simulated. But it would be interesting if the authors would be able to present (also in the discussion section) some future research directions, which can contribute to impact assessement of that fishing habit.

Response: In Section 4.1 of the discussion, we have added some future research directions.

Some spelling mistakes can be found in the text, a strict re-reading and corrections are needed.

Response: We have checked and corrected the spelling mistakes.